# Determining counselling communication strategies associated with successful quits in the National Health Service community pharmacy Stop Smoking programme in East London: a focused ethnography using recorded consultations

Carol Rivas,[1,2] Ratna Sohanpal,[2] Virginia MacNeill,[3] Liz Steed,[2] Elizabeth Edwards,[2] Laurence Antao,[2] Chris Griffiths,[2] Sandra Eldridge,[2] Stephanie Taylor,[2] Robert Walton[2]

¹Social Science Research Unit, University College London, London, UK
²Centre for Primary Care and Public Health, Queen Mary University of London, London, UK
³Department of Public Health, University of Oxford, Oxford, UK

**Correspondence to**
Dr. Carol Rivas;
c.rivas@ucl.ac.uk

## ABSTRACT

**Objectives** To determine communication strategies associated with smoking cessation in the National Health Service community pharmacy Stop Smoking programme.

**Setting** 11 community pharmacies in three inner east London boroughs.

**Participants** 9 stop smoking advisers and 16 pairs of smokers who either quit or did not quit at 4 weeks, matched on gender, ethnicity, age and smoking intensity.

**Method** 1–3 audio-recorded consultations between an adviser and each pair member over 5–6 weeks were analysed using a mixed-method approach. First a content analysis was based on deductive coding drawn from a theme-oriented discourse analysis approach and the Roter Interaction Analysis System. Core themes were identified through this quantification to explore in detail the qualitative differences and similarities between quitters and non-quitters.

**Results** Quantitative analysis revealed advisers used a core set of counselling strategies that privileged the 'voice of medicine' and often omitted explicit motivational interviewing. Smokers tended to quit when these core strategies were augmented by supportive talk, clear permission for smokers to seek additional support from the adviser between consultations, encouragement for smokers to use willpower. The thematic analysis highlighted the choices made by advisers as to which strategies to adopt and the impacts on smokers. The first theme 'Negotiating the smoker–adviser relationship' referred to adviser judgements about the likelihood the smoker would quit. The second theme, 'Roles of the adviser and smoker in the quit attempt', focused on advisers' counselling strategies, while the third theme, 'Smoker and adviser misalignment on reasons for smoking, relapsing and quitting', concerned inconsistencies in the implementation of National Centre for Smoking Cessation and Training recommendations.

## Strengths and limitations of this study

► To our knowledge, this is the first study to consider verbal interactions in real-world consultations between community pharmacy advisers and smokers in the National Health Service community pharmacy Stop Smoking programme.

► Thirty-two smokers from a larger sample were matched on demographic variables to provide 16 matched pairs of 4-week quitters and non-quitters, enhancing confidence in findings.

► The findings may not be transferable to other pharmacies, particularly in less disadvantaged areas or to over-the-counter community pharmacy interactions since the consultations took place in dedicated consultation rooms.

► Our study findings are limited to interactions during the consultation, as appropriate to the methods chosen, and we did not systematically explore previous quit attempts and their influence on results.

**Discussion** Advisers in community pharmacies should use the advantages of their familiarity with smokers to ensure appropriate delivery of patient-centred counselling strategies and reflect on the impact on their counselling of early judgements of smoker success.

## INTRODUCTION

UK community pharmacies have adopted public health and healthy living tasks that were previously the domain of the general practitioner.[1] These 'enhanced services' are embedded in UK health service policy and embraced by pharmaceutical professional bodies.[2–9] Community pharmacies

have great potential to provide such healthcare, particularly for people from disadvantaged groups,[10] given their community setting, long opening hours, accessibility, familiarity and informality.[11–13] Pharmacy-enhanced services therefore provide an opportunity to address health inequalities and may free general practice time for patients with more complex problems.[14]

The National Health Service (NHS) Stop Smoking programme is one of the most frequently delivered enhanced community pharmacy services in England[15 16] and its effectiveness in 2015/2016 measured by smoker self-reported 4-week quit rates was only slightly lower than for general practice (44% vs 49%).[17] But neither general practice nor community pharmacies achieve the suggested optimal quit rate of 70%,[7] and there is considerable variation in rates within the same setting.[17] Moreover, the proportion of smokers enrolling on the Stop Smoking programme in community pharmacies rather than general practitioner (GP) surgeries and other settings has remained stable since the programme began in 2006, falling between 18% and 21%, for example 20% in 2015/2016.[7 17] This is despite government efforts to shift some of the GP workload to them.[7 15]

Enhanced services have necessitated a change in mind set for pharmacy staff with a move towards the person-centred care model currently advocated within health services.[17–26] Pharmacy staff tend to be inadequately trained in this model[24] and may lack necessary consultation skills,[24 26] which may affect the growth of the Stop Smoking service in pharmacies. We therefore used a 'focused ethnography' approach[27 28] to explore counselling strategies used in consultations between pharmacy Stop Smoking advisers and smokers. Our primary aim was to use the findings to develop an intervention intended to improve both service uptake and quit rates in the community pharmacy Stop Smoking programme, as reported elsewhere.[29]

## METHODS
### Focused ethnography
Using focused ethnography[27 28] we collected and analysed recordings of naturalistic (real world) consultations between pharmacy Stop Smoking advisers and smokers; recordings were made by the advisers themselves for the study. We considered verbal interactions that might affect smoker engagement, maintenance within the pharmacy programme and success in quitting.

In traditional ethnographic work, researchers immerse themselves in the social worlds of their participants to better understand the ways they interact with these worlds. They then generate textual or visual accounts of this for dissemination.[30] In focused ethnography, field visits are either much shorter and selectively focused or replaced by audio, video or photo recordings which may be undertaken with the researcher absent from the scene, as in our study. This approach is useful for answering a specific research question when traditional ethnography is impractical.[28]

### Pharmacies and advisers
Consultations were recorded between November 2013 and May 2014 at community pharmacies contracted by Public Health England to provide the Stop Smoking programme in three inner East London boroughs (Newham, Tower Hamlets, City and Hackney). These areas are economically disadvantaged compared with other areas of London and with England as a whole[31] and have relatively poor NHS Stop Smoking programme attendance.[13 15 16] We only included community pharmacy chains or independent single pharmacies, which are distinct from those in institutional settings such as NHS hospitals. Consultations had to take place in a private room. Only the adviser and smoker were present though sometimes the consultations were interrupted as they often took place in rooms that doubled as store rooms, which other staff needed to access.

To select pharmacies, we used maximum variation sampling by borough, pharmacy size and length of time on the Stop Smoking programme. Within pharmacies, we invited any staff members certificated through the National Centre for Smoking Cessation and Training (NCSCT) programme, with a mixture of pharmacists and counter assistants therefore included in the study. We considered sampling for variation in years spent counselling for smoking cessation and in certification level in the NCSCT, but there was too much missing data for this to be useful. In any case, our focus was on the interaction between smoker and adviser per se. We visited pharmacies on spec or by appointment, explained the study and if relevant took consent and trained them in research methods and governance, so that they could take smoker service user consent and record, securely store and transfer their own consultations to the research team. This meant the natural rhythm of the service was not disrupted. Ethical approval for the study was obtained from the National Research Ethics Service Committee South Central, Berkshire B (reference number 13/SC/0189).

### Smokers
All adult smokers recruited onto the NHS Stop Smoking programme during the course of the study were eligible and invited to join the study by the pharmacy adviser. Smokers were designated as having quit if they self-reported a smoke-free status at 4 weeks after their set quit date and provided an adviser with an expired carbon monoxide (CO) reading of less than seven parts per million using a CO monitor. Smokers lost to follow-up at 4 weeks were considered still smoking following the Russell criteria.[32]

### Data collection
Community pharmacy NHS Stop Smoking programme consultations and prescriptions for medication may

continue for up to 12 weeks with formal consultations weekly or 2 weekly and informal drop-ins between these times as needed.[7 33] The core period for the programme ends at 5 to 6 weeks which tends to equate with the fourth week after the quit date, usually set at the first consultation.[7 33]

In line with our focused ethnography approach, Stop Smoking advisers consented eligible smokers and selectively audiorecorded:

► Their first consultation, which tends to set the parameters for the remaining sessions, includes the smoker setting a proximal quit date and is usually up to three times as long as subsequent sessions (averaging 15 min vs 5 min)[34];

► The consultation at 2 weeks post proposed quit date where the adviser explores and attempts to solve any issues in the quit attempt;

► The consultation at 4 weeks post proposed quit date at which the smoker's quit status is formally designated (as the 4-week quit status) for Public Health England audit statistics and pharmacy remuneration[35] (see Sohanpal *et al*[36] for a discussion of remuneration in relation to our larger study).

We call these weeks 1, 2 and 4 for simplicity. Recordings were anonymised and transcribed as they were received. Pseudoanonymised metadata (such as demographic data and smoking intensity for the smokers and training and demographic data for the advisers) were entered with anonymised recordings and transcripts onto a secure clinical trials database.

### Matched pairs

We refer to quitters and non-quitters throughout this paper, but in consultations 1 and 2 this is a retrospective designation for eventual quitters and non-quitters. We matched 16 pairs of 4-week quitters and non-quitters (ie, 32 smokers) on gender, ethnicity, age (according to UK Office for National Statistics [ONS] age bands) and

smoking intensity (fewer than 10 cigarettes a day, 10–20 a day, more than 20 a day). We had initially aimed for 20 matched pairs but obtained saturation at this lower number and so stopped data collection at this point.

### Consultations analysis

We used a mixed-methods analysis approach, in which themes were identified deductively and inductively and then quantified (figure 1). First, we developed deductive themes from a list of: the key linguistic devices used in theme-oriented discourse analysis,[37] which considers how language use constructs professional practice; and the structuring of the medical consultation in the Roter Interaction Analysis System (RIAS),[38] which is a widely used method for coding and quantitatively analysing the conversational strategies used structurally in medical consultations. Our approach thus has its origins in psycholinguistics and its purpose is to look at what talk itself is doing as constitutive of social action rather than as representational of inner psychological states, behaviours, beliefs or attitudes.[37]

The core team then applied the deductive themes to the consultation transcripts and also looked for emergent (inductive) themes,[39 40] immersing themselves in the data, reading and rereading each consultation transcript and meeting to discuss themes. CR led development of themes from the data. The team constantly compared deductive and emerging inductive themes with each other and also compared data within themes to ensure each emergent theme was discrete from all other themes and representative of the relevant data.[39 40] CR operationalised the themes, for a second researcher (RS) to code 20% of the data independently. Differences were reconciled by discussion, until a Cohen's kappa statistic of 76% was obtained.

Once themes were determined, a quantitative summary of these was developed. First, for each theme we quantified the number of smokers whose consultations included the

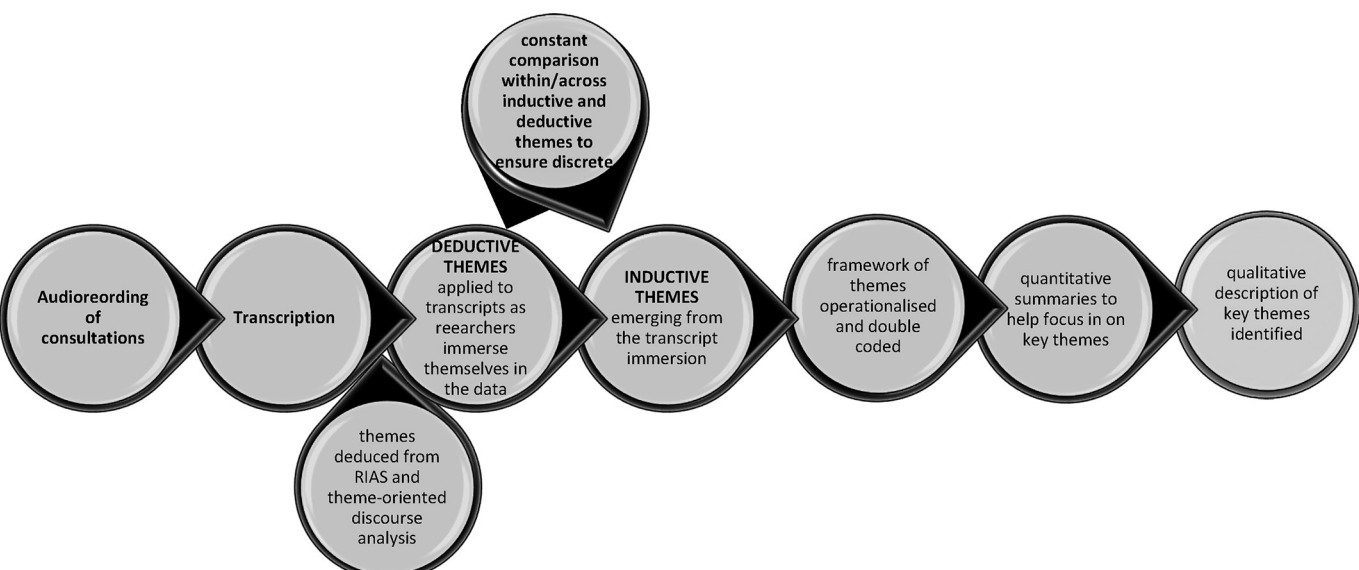

**Figure 1** Analysis process. RIAS, Roter Interaction Analysis System.

theme (A of table 1). This suggested what strategies were typically used in the counselling sessions (thus a core set of strategies). We also calculated the average frequency of use of a theme within consultations in which it was mentioned (D of table 1) by dividing the total number of occurrences of a theme across all consultations (column C of table 1) by the number of consultations in which it occurred (column B of table 1). We only counted one occurrence in each 'turn' the smoker or adviser took to talk (where a conversation is made up of lots of turns at talking). This gave us a measure of the relative importance of each theme; a theme with an average frequency of 1 may be considered less important overall than a theme with an average frequency of 4, that is, it may be used only in very specific circumstances or idiosyncratically. To obtain the frequency measure, we divided by the number of consultations with a mention of the theme rather than all consultations because we wanted a measure that indicated repetitions. We could have provided an alternative measure of the relative bias towards a theme within consultations by dividing the number of mentions of all themes in each consultation by the number of mentions of each specific theme. However, this measure would be sensitive to the absence of other themes because they were not relevant (for example because the 3 weeks we recorded had different foci) or because they had been covered outside of the recording. For similar reasons we did not quantify the number of words used for each theme.

With all quantitative measures (as with the qualitative analysis) we looked for similarities and noted any differences between eventual quitters and non-quitters. We used the quantitative measures to give us a broad picture of the consultations and themes so that we could determine which to explore in more detail qualitatively. We focused on themes that were associated with quit success or lack of success and that were of high or moderate frequency of use across consultations or that appeared to be core to the sessions. We also considered strategies that were rarely used but that might be important for service improvement. This led to a list of key themes, which we report on in more detail in the next section. Given the small numbers involved, tests of significance are not reported here.

## FINDINGS
### Participants
Thirty-nine per cent (11/28) of recruited advisers provided 158 audiorecordings from 53 smokers; two further pharmacies dropped out due to lack of time. Considering only the 16 matched pairs, these were seen by 9 of the 11 advisers, with a median of 3 smokers each (range 1–7). Table 2 compares baseline characteristics of the matched pairs to the total sample. While women were under-represented in the matched pairs, other characteristics related to cessation outcome were broadly similar. This includes medication use; though we did not monitor

for this systematically, the transcripts indicate that similar numbers of quitters and non-quitters were on vareniciline, for example. Most advisers in the total group were male (89%) and from pharmacies in City and Hackney (89%). Sixty-seven per cent of advisers were Indian, and 11% were from other black and minority ethnic groups, with missing ethnicity data for one. Importantly, findings were similar whether the adviser was a pharmacist or trained counter assistant. With one exception, there was no indication of a lesser authority in counter assistants since they, like the pharmacists, had received special training in smoking cessation counselling. The exception had only recently been trained and was being shadowed by the pharmacist. The small number of advisers in the study precluded analysis of the effect of other variables on the consultation. In any case our analysis was focused on what talk itself does and was not intended to explore these effects.

### Themes
Thematic analysis reached saturation and revealed three overarching themes in the data:
► Negotiating the smoker–adviser relationship;
► The roles of the adviser and the smoker in the quit attempt;
► Smoker and adviser misalignment on reasons for smoking, relapsing and quitting.

These were made up primarily of the subthemes developed deductively from psycholinguistics as described in the methods. Table 2 provides a quantitative overview of the themes and selected subthemes based primarily on the RIAS and theme-oriented discourse analysis, which we focus on in this paper, and compares quitters and non-quitters. We follow this with a narrative description of themes; extracts are identified by smoker (P), adviser (A) and consultation week (W) while the quantifiers most, common or many, occasionally or some and a few or rarely refer to a theme occurrence of 75%+, 50%–74%, 25%–49% and less than 25%. Identification numbers used by us to manage the data are also used to attribute extracts from the data in the narrative text. The first consultation should consider smoker tobacco use and motivation to quit (some advisers used a questionnaire type script) and subsequent ones build on this. Here, we focus on what the talk does within any adviser–smoker interaction and the association of interaction strategies per se with quitting or not, hence aggregating the data across consultations, while mindful of which consultations they represent. We consider the data by consultation sequence in further analyses to be reported elsewhere, which support the findings of this paper.

From the quantitative data, we determined that the core set of strategies used by advisers (occurring in 50% or more of smokers) were praise, biomedical talk and advice, mention of the side effects of medication, collaborative talk, the use of props and strategies to cope with the urge to smoke and the use of monitoring and surveillance talk. Significantly, explicit use of motivational talk is not

**Table 1** Quantitative comparisons between quitters and non-quitters

| Theme* | A: Number of smokers with theme | | B: Number of consultations with theme (%) | | C: Total number of occurrences overall Quitters, non-quitters | D: Mean frequency of mentions in consultations in which theme appears (ie, columns C/B) Quitters, non-quitters |
|---|---|---|---|---|---|---|
| | Number of quitters with theme (% out of n=16) | Number of non-quitters with theme (% out of n=16) | Quitters out of 30 consultations, n (%) | Non-quitters out of 29 consultations n (%) | | |
| Negotiating the smoker–adviser relationship | | | | | | |
| Lifeworld talk | 10 (63) | 9 (56) | 19 (63) | 18 (62) | 75, 45 | 3.95, 2.5 |
| Adviser being non-judgemental | 4 (25) | 3 (19) | 6 (20) | 5 (17) | 9, 10 | 1.50, 2.00 |
| Adviser praise | 10 (63) | 9 (56) | 22 (73) | 20 (69) | 64, 53 | 2.91, 2.65 |
| The roles of the adviser and the smoker in the quit attempt | | | | | | |
| Receiving biomedical information and advice | 14 (88) | 11 (69) | 25 (83) | 16 (55) | 106, 48 | 4.24, 3.00 |
| Importance of support | 8 (50) | 5 (31) | 14 (47) | 6 (21) | 31, 13 | 2.21, 2.17 |
| Mentioning side effects of medication | 9 (56) | 8 (50) | 12 (40) | 11 (38) | 34, 26 | 2.83, 2.36 |
| Adviser mentioning relapse | 4 (25) | 3 (19) | 9 (30) | 6 (21) | 12, 6 | 1.33, 1.00 |
| Motivational talk | 5 (31) | 5 (31) | 15 (50) | 12 (41) | 37, 34 | 2.47, 2.83 |
| Confidence in being able to quit | 3 (19) | 1 (6) | 8 (27) | 2 (7) | 8, 3 | 1.00, 1.50 |
| Need for willpower | 8 (50) | 7 (44) | 9 (30) | 13 (45) | 17, 18 | 1.89, 1.38 |
| Collaborative talk (eg, 'we') | 10 (63) | 8 (50) | 20 (67) | 15 (52) | 67, 51 | 3.35, 3.40 |
| Ownership of the quit | 5 (31) | 4 (24) | 8 (27) | 5 (17) | 16, 6 | 2.00, 1.20 |
| Managing smoker expectations of the programme | 5 (31) | 9 (56) | 9 (30) | 11 (38) | 19, 30 | 2.11, 2.73 |
| The use of quit props and strategies | 11 (69) | 10 (63) | 22 (73) | 22 (76) | 73, 39 | 3.32, 1.77 |
| 'Open-door' talk (invitations to informal consultations between the formal ones) | 9 (56) | 7 (44) | 10 (33) | 8 (28) | 15, 12 | 1.50, 1.50 |
| Monitoring and surveillance | 14 (88) | 15 (94) | 21 (70) | 17 (59) | 38, 38 | 1.81, 2.24 |
| Smoker and adviser misalignment on reasons for smoking, relapsing and quitting | | | | | | |
| Smoker explanations for smoking | | | | | | |
| Stress | 6 (38) | 9 (56) | 8 (27) | 13 (45) | 19, 17 | 2.38, 1.31 |
| Social factors | 4 (25) | 3 (19) | 6 (20) | 4 (14) | 8, 6 | 1.33, 1.50 |
| Financial reasons | 5 (31) | 5 (31) | 5 (17) | 6 (21) | 6, 8 | 1.20, 1.33 |
| Health problem | 8 (50) | 11 (69) | 8 (27) | 13 (45) | 19, 32 | 2.38, 2.46 |
| Smoker explanations for wanting to quit | | | | | | |
| Quitting for families (including children) | 5 (31) | 3 (19) | 6 (20) | 3 (10) | 15, 3 | 2.50, 1.00 |
| Obligations to self and others | 2 (13) | 1 (6) | 4 (13) | 1 (3) | 6, 1 | 1.50, 1.00 |
| Negative feelings towards smoking | 1 (6) | 3 (19) | 1 (3) | 5 (17) | 1, 5 | 1.00, 1.00 |

Continued

**Table 1** Continued

| Theme* | A: Number of smokers with theme | | B: Number of consultations with theme (%) | | C: Total number of occurrences overall Quitters, non-quitters | D: Mean frequency of mentions in consultations in which theme appears (ie, columns C/B) Quitters, non-quitters |
|---|---|---|---|---|---|---|
| | Number of quitters with theme (% out of n=16) | Number of non-quitters with theme (% out of n=16) | Quitters out of 30 consultations, n (%) | Non-quitters out of 29 consultations n (%) | | |
| Negative impact of smoking on appearance (eg, Ageing effects) or identity (eg, Smelly) | 1 (6) | 3 (19) | 2 (7) | 5 (17) | 2, 8 | 1.00, 1.60 |
| Adviser-suggested motivator for smoker to quit | | | | | | |
| Financial | 4 (25) | 6 (38) | 10 (33) | 8 (28) | 14, 9 | 1.40, 1.13 |
| Health scare tactics | 4 (25) | 3 (19) | 5 (17) | 3 (10) | 8, 5 | 1.60, 1.67 |

*Themes shown here have been selected from a larger pool for this paper.
(A) number of quitters and non-quitters for which each theme was identified (second and third columns); (B) number of consultations overall in which each theme was identified (middle columns); (C) total number of occurrences in all consultations (penultimate column); (D) relative frequency with which themes were used in individual consultations (final column). In the narrative text we only consider values for A and D.

included in this core set.[41] Further strategies were added to the core set in eventual quitters: talk on the importance of support; open door talk and the need for willpower. Managing expectations formed part of the core set for non-quitters only. These themes are discussed in more detail below.

### Negotiating the smoker–adviser relationship

Lifeworld talk was common across the groups, though occurring in slightly more eventual quitters and with more occurrences within their consultations. This is talk about a person's experience of life with all its contextual and situational nuances.[42–45] Through lifeworld talk, the advisers gained a shared understanding with the smoker of the context of the smoker's quit attempt and the individual difficulties they faced. The adviser could then discuss approaches to quitting that took into account such things as the stress that the smoker in the following extract describes.

A: So while you're on Champix and you left work and you're a bit stressed did you say?

S: Yeah, because there's many changes in the work you remember when you told me in the beginning, like try to, don't change the work and the things. But they change our store manager, my manager is leaving. The work we used to do, six persons now going to be in two persons, oh my god. It's just,

A: So you're a bit more stressed at work.

S: Terrible. It's terrible.

A: So you came home and you're a bit stressed still, so you felt like you wanted a cigarette? But you didn't have a cigarette? (P807-A81-W4, quit)

The advisers made it clear to around one quarter of smokers that they were not going to judge them for any lapses and failures, as a way to motivate them to continue the programme, illustrated in the following extract. This is in line with good practice.

A: Fantastic. So again, my idea is to mentor you and support you and that's what it's about, yeah? I'm not here to tell you off. It's about where you're finding it hard and what we need to work together to find a solution for you. (P107-A1-W1, quit)

Praise was common across groups, but with marked qualitative differences. Quitters received limited and brief praise. By contrast, eventual non-quitters were given extensive (extreme formulation[46 47]) praise even when they had not achieved the NCSCT-recommended 'not a puff', as below.

S: Well, it's actually helped a lot, so I'm only smoking around ten.

A: Ten plus? Ten to fifteen?

S: Ten plus.

A: That's fantastic. In the space of six, 7 days you've cut down by over half. (P4-A1-W2, not quit)

**Table 2** Smoker demographic and smoking data summaries for total number sampled and for the matched pair groups

| Variable | Quitters (16) | Non-quitters (16; 6 LTF) | All (53; 9 LTF) |
|---|---|---|---|
| Female | 3 (19%) | 3 (19%) | 18 (34%) |
| Age (median, range) | 47 (26–58) | 41 (28–59) | 41.13 (18-67) |
| Occupation | | | |
| Managerial and professional | 2 (13%) | 0 | 5 (9%) |
| Unemployed | 10 (63%) | 10 (63%) | 25 (47%) |
| Routine manual | 3 (19%) | 3 (19%) | 8 (15%) |
| Student | 0 | 0 | 1 (2%) |
| Intermediate | 0 | 2 (13%) | 7 (13%) |
| Retired | 0 | 0 | 1 (2%) |
| No answer | 1 (6%) | 4 (25%) | 6 (11%) |
| Ethnicity | | | |
| White British | 11 (69%) | 9 (56%) | 30 (57%) |
| South Asian | 3 (19%) | 3 (19%) | 8 (15%) |
| Other white | 2 (13%) | 3 (19%) | 10 (19%) |
| Black African | 0 | 0 | 1 (2%) |
| Black Caribbean | 0 | 0 | 1 (2%) |
| Missing data | 0 | 1 (6%) | 2 (4%) |
| Cigarettes smoked | | | |
| 40 | 1 (6%) | 0 | 1 (2%) |
| 30 | 2 (13%) | 2 (13%) | 5 (9%) |
| 25 | 3 (19%) | 3 (19%) | 8 (15%) |
| 20 | 7 (44%) | 9 (56%) | 25 (47%) |
| 15 | 1 (6%) | 2 (13%) | 5 (9%) |
| <15 | 2 (13%) | 1 (6%) | 9 (17%) |
| London Borough | | | |
| Tower Hamlets | 2 (13%) | 5 | 12 (23%) |
| City and Hackney | 14 (88%) | 10 | 38 (72%) |
| Newham | 0 | 0 | 3 (6%) |

LTF, lost to follow-up.

## Roles of the adviser and the smoker in the quit attempt

Biomedical information and advice was imparted across the groups but quitter consultations were most heavy with it. This may mean they were more open to biomedical talk or that the adviser got an impression that they were or it may be an incidental finding. Side effects of the medication and withdrawal symptoms from stopping smoking, including weight gain, were discussed with many smokers. The possibility of relapse was discussed with only 25% of quitters and 19% of non-quitters. There was some compensation in the degree to which the use of strategies to deal with cravings (hence risk of relapse) was commonly encouraged (see extract later in this section).

Advisers made attempts to manage many of the eventual non-quitters' expectations of the programme as well as emphasising their required commitment. Both strategies were undertaken only occasionally with quitters. Many quitters but only some non-quitters were told of the importance of advisers' counselling support. Advisers' motivational talk was occasional in both groups and did not form part of the core set of strategies and advisers only rarely asked for (and therefore got) expressions of confidence in being able to quit (three quitters over eight consultations vs one non-quitter over two consultations), a specific element of motivational interviewing. The need for the smoker to draw on their own willpower to quit was considered in most quitters but only some non-quitters, and the necessary balance between adviser support and more personal willpower or motivation was only discussed with one non-quitter (in the following extract) and one quitter.

S: I've got a photograph of me taken when I was in intensive care in hospital, I should take a look at that every time I want a cigarette.

A: Perfect! You need some motivation.

S: I don't want to end up like that again.

A: Exactly, you need that motivation. And if there's anyone else that can help you also…

S: My mum, yeah, she's very supportive.

A: Sure. You need that support as well. But you've got us as well because you'll be coming here every week. (P806-A81-W1, not quit)

Collaborative talk (such as the use of the phrase 'we will succeed'), a supportive stratagem, was common in both groups. The related topic 'smoker's ownership of the quit' was rarely mentioned in non-quitter consultations and only slightly more so in quitters'. Some messages were implied rather than explicit. The use of props and strategies to deal with cravings requires some ownership and self-management of the quit and was recommended at least once in a majority of smokers and consultations, though more frequently discussed within quitter consultations. Many different props and strategies were suggested, with the following only one example.

A: I want you to make sure you start off in the morning; you've got your bottle of one point five litres to two litres. Your aim, you start in the morning, you finish at the end of the day. Every time you want a cigarette, you're going to drink the water. You're going to complete it. (P101-A11-W1, not quit).

The advisers invited many smokers to drop into the pharmacy for support and advice any time the shop was open, a particular advantage that pharmacies have over other healthcare settings. Such 'open door' talk, offering a more informal variant of adviser support, was said to fewer eventual non-quitters and was less successful for these.

In most smokers, the CO monitor was depicted by the adviser as a monitoring or surveillance tool that was used to check the smoker's progress and even catch the smoker out:

A: Well, it's good that you're here, we will help you in any way we can. What I ask you every week is if I can take a reading from you. This is called a CO monitor and this machine measures the amount of carbon monoxide inside of your system. And I call this my lie detector. I'm not going to say that you're going to lie to me, but if you do come in and I say have you had any cigarettes, and you say no, I haven't had any, this will tell me the truth. (P7-A1-W1, quit)

### Smoker and adviser misalignment on reasons for smoking, relapsing and quitting

An actual health problem or worry about getting one was cited as a reason to quit by more than half the smokers and was invoked slightly more commonly by non-quitters. There was an inverse relationship between this and adviser use of health scare tactics to motivate the smoker, though these tactics were relatively uncommon. Financial reasons for wanting to quit were cited by just under one-third of smokers. However, the advisers tried to motivate the smokers with talk of financial savings on 23 separate occasions. Initial misalignment is demonstrated in the following extract for example; though there is evidence of subsequent 'repair' of this at the end of the extract, our focus in the current analysis was on whether misalignment arose at all.

S: I mean I don't want to die. I don't want to have a stroke.

A: So that's your main impulse now and more likely you'll succeed because of that.

S: I don't want to end up like that…

A: The other reasons why maybe you feel I'll give up now, maybe save a bit of money or whatever before…

S: That's the backburner, that's nothing to do…

A: Exactly. Was that the reasons before?

S: No.

A: What were the reasons before that you wanted to stop smoking?

S: Smell; I don't like the smell. I've got this thing, when I eat, people smoke around me, I don''t like it, I don't like the smell of it when I'm eating. People light one up when you're having something to eat and oh, it's disgusting, I don't like the smell. (P806-A81-W1, not quit)

Quitters were twice as likely to mention their families (including children) as motivation though under a third did so. Other reasons for smoking were cited relatively infrequently. Notably the misalignment or mismatch between smoker declaration of health as motivation to quit and adviser comments on health problems caused by smoking was more likely in non-quitters, and advisers were also less likely to mention motivational factors at all to eventual non-quitters. In quitters' consultations, advisers were likely to mention motivators, especially financial, even when the smoker did not.

## DISCUSSION

We applied rigorous and systematic analytical methods to a sample of audiorecordings of real-world community pharmacy adviser–smoker NHS Stop Smoking consultations with the premise that the community pharmacy Stop Smoking programme should be person-centred. Specifically, we explored associations of thematic patterns in the verbal interactions with 4-week quit success. Although there are a number of ethnographic studies of client–pharmacy staff communication processes or interactions,[24 38 48–71] there is none on stopping smoking in the community. Our overall findings accord with those of Pilnick's study[49–51] in a UK hospital paediatric outpatient clinic pharmacy, the most comparable study to ours, which focused on advice giving in the counselling role, but we provide rich new data.

There were some clinically potentially significant omissions across the data. For example, talk about relapse was

rare, though there was some compensation in the degree to which the use of strategies to deal with cravings was frequently encouraged and the biomedical problems of quitting often explored. We found many examples of good practice, including significant evidence of life-world talk,[42 43] which was common across the two groups. Mishler,[43] applying the Habermas theory of Communicative Action[72] to medical encounters, itself derived from Husserl's phenomenological concept of consciousness and subjectivity,[44] showed how this enables person-centred support. It provides the pharmacy advisers with the smokers' own nuanced and contextualised[73–81] accounts of their personal daily lives, in which they manage their quit attempts. When the adviser uses these to shape their recommendations, advice and medical support, adviser consultations are more likely to be effective.[44 82] The occurrence of lifeworld talk in our data suggests that the community setting and the advisers' social familiarity[81] with the smokers as customers enable such smoker-centred care, in a way that is potentially greater than is possible in GP consultations. This supports other research suggesting that community pharmacies are ideally placed for smoking cessation and other health behaviour change tasks.[10–13]

Nonetheless, the core set of strategies common across both quitter and non-quitter groups focused on the practical and the medical, biomedical talk and advice (the 'voice of medicine').[42 43] This was certainly an important part of consultations and appropriate given the pharmacy staff member's role as adviser and the need for such talk in the first consultation in particular. But unlike biomedical talk, lifeworld talk was not included in this core set, in other words, it was less consistently used. We found, from more detailed analysis, that differences between smokers in the way lifeworld talk was used could be related to quit success or failure.

It appeared that advisers used lifeworld talk to make judgements about likely smoker quit success. This then compromised its effective use. We found this affected non-quitters disproportionately, and this seemed due to the adviser focussing on possible problems in those they judged as less likely to quit and making assumptions about reasons the smoker found it hard to quit. This counselling approach was not used in those who went on to quit. Our data show that in this way, prejudgements led to three sources of misalignment with, or confusion in, the smokers who did not quit. First, advisers frequently offered casual 'open-door' drop-in support between consultations to smokers. But these offers were less clear or explicit in their purpose when made to non-quitters and appear to have simply confused them as to when they were meant to attend. Second, praise and willpower talk were common in both quitters and non-quitters, but the way they were communicated was often poorly matched to smoker effort. Extreme statements made in praise of non-quitters when they were still smoking created cognitive dissonance. Third, the advisers drew on financial and health benefits as motivators for quitting in those

who then failed to quit and had cited other reasons for wanting to quit. The misalignment between the smoker's reasons for smoking or wanting to quit and the adviser's perceptions of what these were would result in the adviser setting inappropriate goals for the smoker.[72–82]

These misalignments may therefore explain why a smoker's sharing of their lifeworld was not always associated with quit success. This interpretation of the data is supported by similar findings in our separate analysis of adviser interviews, with some advisers explicitly stating that they emphasised particular strategies in those smokers they considered would find it harder to quit, to try to help them.[36] Misalignments push interactants 'out of sync' with each other, making it hard for them to achieve a common goal. Nyugen considers how misalignments in pharmacy counselling need to be delicately managed not to affect the tenor of the rest of the conversation.[62] Misalignments were noted by Pilnick[49–51] as problematising the adviser's advice giving, but we have shown how they develop to create issues in the adviser's cognitive development of what that advice might be as well as how their counselling support should be configured.

Our data also show that the advisers drew on sometimes apparently contradictory strategies to induce a quit. For example, they often combined reassurance of a non-judgemental approach with the use of the CO monitor as a surveillance tool. While this seems to have been successful, biomedical talk and advice, talk of being non-judgemental, monitoring and surveillance and the advisers' use of authority and expertise to prejudge likely smoker success are all premised on a power differential between adviser and smoker and may therefore appear to sit in dialectical tension with the development of shared understandings and social familiarity from the community setting. Social familiarity and power differentials are not mutually exclusive but they are less likely to coexist the greater the difference in social status between two interactants.[72 78 81 83 84] Thus, the pharmacy community setting provides particular communication issues that are less likely to occur in encounters with doctors and medical specialists. Social familiarity is shown in our data to have the potential to enhance quit success and power differentials to compromise this, and the best outcomes might be achieved when advisers strike a balance between the two, optimising the benefits of the community pharmacy setting. Figure 2 demonstrates this. Considering recommendations for pharmacy adviser smoking cessation care, we found successful quits to be more likely when there was more smoker-centred care with counselling plans that were based on the smokers' own lifeworlds and that matched strategies such as praise to the smokers' actual rather than predicted behaviours.

## Strengths and limitations

Our study contributes to current knowledge and practice with novel data from a pharmacy service that has not

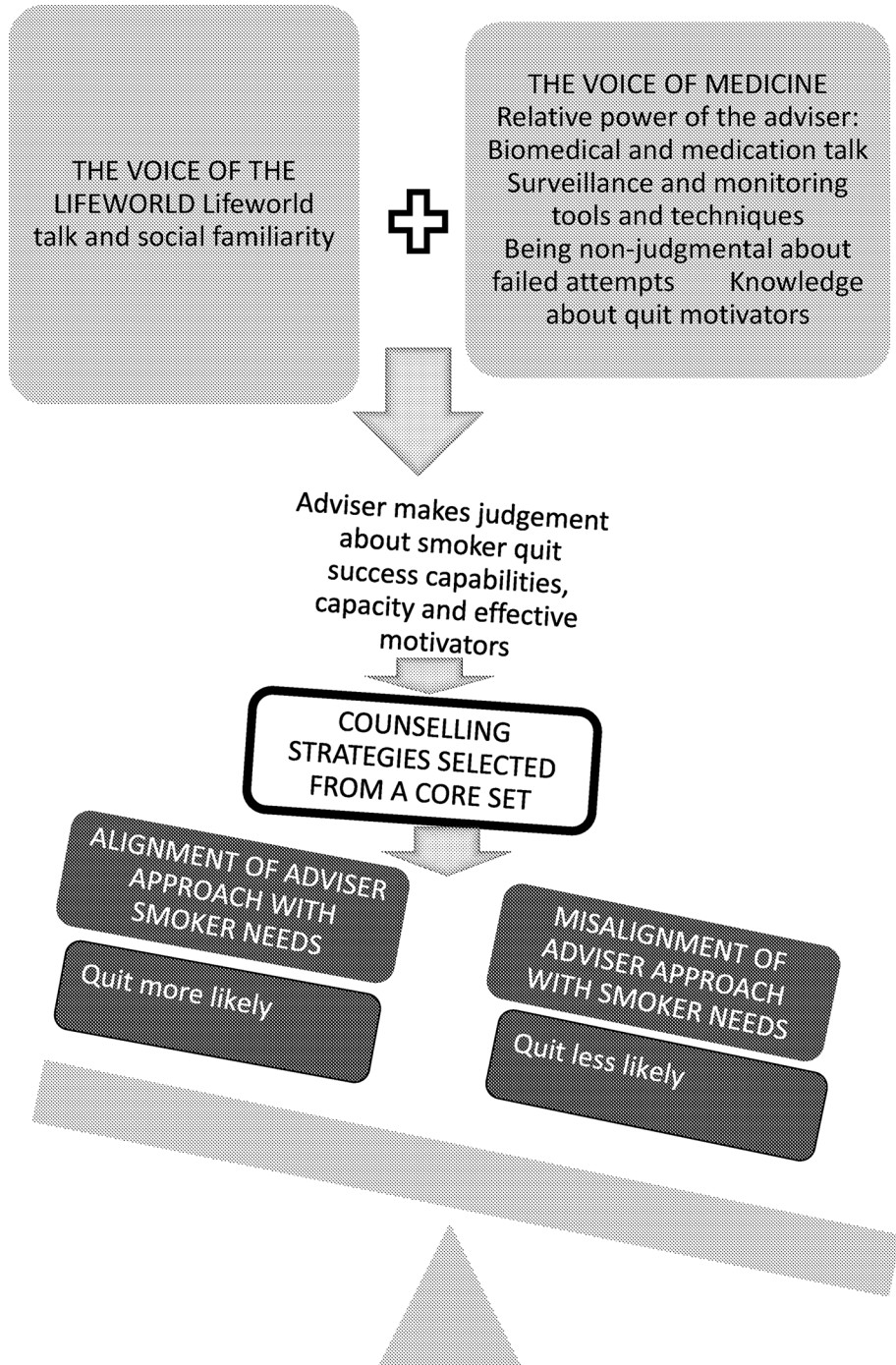

**Figure 2** Tentative model of misalignment in the community pharmacy NHS Stop Smoking service consultation. NHS, National Health Service.

previously been subject to ethnographic study. As this was a qualitative study, we considered data from a small and varied sample of participating staff members, smokers and community pharmacies, adding to richness of the findings. Since we were comparing two groups, we matched them on relevant variables to increase the dependability of our analysis. We achieved theme saturation within this matched pair design, with adequate agreement between coders, group data sessions and consideration of negative cases in the data adding to credibility and dependability of findings. Themes reported in this analysis correspond with some of the themes developed separately from semistructured qualitative interviews with advisers and smokers from the same study.[36] Our study has enabled us to identify key elements of talk and can be used to strengthen consultations and make them more effective.

Our study also has limitations. We were not successful in recruiting pharmacy chains and make no claims to transferability of the findings to other pharmacies, particularly those in more affluent areas. Further, it may not be possible to extrapolate findings from these consultations, which were undertaken in a private room, to over the

counter community pharmacy interactions. In collecting naturalistic qualitative data, our aim was rather to provide in-depth exploration of communication practices. This was usefully achieved. A number of contextual and demographic variables are known to affect pharmacist–smoker communication and quit successes.[85] We were unable to explore these in our small sample. We did not case match by adviser ethnicity or other adviser variables as this would have required oversampling for our specific research question, which was not intended to explore this, but this might be a topic for different research. However, we systematically looked for adviser-specific patterns and found that our analysis held irrespective of adviser. Participants were not blinded to the nature of the research and advisers had a potential vested interest in successful quits, creating a possible performance effect. However the unobtrusive data collection method will have reduced this risk with smokers and the effect would have been the same for quitters and non-quitters. There may have been an intrinsic difference between the quitters and non-quitters before they even entered the consultation room, and our study was not designed to explore this. For example, although baseline characteristics known to be related to quitting, such as gender and age, were taken into account, we did not systematically collect data on other potentially significant variables such as number of previous quit attempts or medication use. Our thematic analysis suggests this would have been useful. The quitters may have been successful irrespective of consultation features; the availability of support itself may be sufficient.[41 85]

## Clinical implications

Advisers had all received NCSCT certification; this training focuses on counselling skills but does not consider implementation in practice. Nor does it cover initial smoker engagement and retention in the service. Moreover, while the NCSCT programme has been shown to be effective,[86] we previously identified through semi-structured interviews that advisers lacked confidence when implementing the training in practice.[36] They wished for more support that built on and reinforced many of the behaviour change techniques learnt through the NCSCT scheme and that also addressed client-centred consultation skills.[36] These issues may partly explain the advisers' inconsistencies in the delivery of and content of the consultations. To improve their practice, advisers might reflect on their use of the considerable lifeworld talk generated by smokers through their social familiarity with the pharmacy advisers. Lifeworld talk should be used to modify adviser counselling to suit key external psychosocial factors, namely the context within which the quit attempt is made and the related motivators to quitting. Our data suggest that currently advisers may instead modify the consultation to suit their assumptions about internal psychological factors such as the smoker's capacity to quit or motivation for quitting.[41] In this way, advisers sometimes develop the consultation in ways that, while well meaningly intended, run the risk of being counter-productive. For smokers judged by them as less likely to quit, this may include inappropriately overemphasising some support strategies (such as praise, managing expectations) and underplaying others (such as the need for willpower and the importance of adviser support). Our data illustrate the potential for misalignment that this creates. Moreover, the exaggerated praise we saw in non-quitter reductions in smoking conflicts with the 'not a puff' rule recommended by the NCSCT. Advisers should use findings from our study to clarify where they might usefully capitalise on the social closeness they seem to share with the smokers they see, with more appropriate use of lifeworld talk and more use of smoker-declared motivators during motivational interviewing.

## Recommendations for research

Our study only considers short-term outcomes, something we are addressing with further research. The analyses have been used to inform the development and content of our training intervention for pharmacy Stop Smoking advisers, evaluated in a cluster randomised trial (grant number RP-PG-0609–10181). This intervention complements NCSCT training and is not intended to replace it. We are currently exploring the microdetail of the talk in the consultations using conversation analysis with a view to further dissemination and use. Our study could be replicated in less disadvantaged areas or in specific minority groups.

**Acknowledgements** The authors thank participants and their community pharmacies. This paper presents independent research funded by the National Institute for Health Research (NIHR) under itsProgramme Grants for Applied Research Programme. The views expressed are those of the authors and not necessarily those of the NHS, the NIHR or the Department of Health.

**Contributors** RW conceived the study. CR undertook the analysis and wrote the manuscript. RS, LA and VM were involved in data collection. RS, LS and RW contributed to the data analysis and interpretation. VM, LS, LA, EE, CG, ST, SE and RW were involved in revising the article critically and contributing important intellectual content. RW is the guarantor.

**Funding** NIHR Programme grant RP-PG-0609-10181.

**Competing interests** None declared.

**Ethics approval** NRES Committee South Central - Berkshire B (reference number: 13/SC/0189.

**Provenance and peer review** Not commissioned; externally peer reviewed.

**Data sharing statement** This study analyses qualitative data and the advisers and smokers did not consent to have their full consultation transcripts made publicly available. They consented for their data to be stored for 5 years as part of the 5-year programme grant project and to be used in this or other studies directly linked to the 5-year project. Supporting excerpts from the raw data (quotes from consultations) are available within the text of the paper. Full transcripts (with identifying information removed) will be available, following completion of the project, on request from the study guarantor, Robert Walton, r.walton@qmul.ac.uk.

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
