## [Reviewer comments · BMJ Open]

ARTICLE DETAILS

TITLE (PROVISIONAL)	Determining counselling communication strategies associated with successful quits in the National Health Service community pharmacy Stop Smoking programme in East London: a focused ethnography using recorded consultations
AUTHORS	Rivas, Carol; Sohanpal, Ratna; MacNeill, Virginia; Steed, Liz; Edwards, Elizabeth; Antao, Laurence; Griffiths, Chris; Eldridge, Sandra; Taylor, Stephanie; Walton, Robert

VERSION 1 – REVIEW

REVIEWER	Afonso Cavaco Faculty of Pharmacy, University of Lisbon. Portugal.
REVIEW RETURNED	18-Jan-2017

GENERAL COMMENTS	I would like to thank the BMJ Open and the authors for the manuscripts entitled “Defining effective communication strategies in the NHS community pharmacy stop smoking programme: an ethnographic study of consultations with smokers”. This is an important area of research for pharmacy practice and public health policies. My few comments are as follows. Abstract Overall, the abstract is dense and somehow difficult for non-native English readers. In Methods, I would suggest to add a mix-methods approach. Maybe a theoretical backbone for informing thematic analysis would help the reader to locate how the main themes were developed. In Results, and following the methodological description, I would start by presenting first the quali results. If content analysis was done 1st, then I would recommend a different wording in Methods. There is a missing 1) item, as 2) and 3) exist. Although sample pairing is mentioned, it was not clear to me if differences were aimed and found. Strengths & Limitations Page 3, Line 11. Theme saturation is a methodological requirement for baseline quality in qualitative research. I would recommend to drop this from the 2nd bullet point. Introduction Page 2, Lines 43-49. The last paragraph is an important description
--

of the work accomplished. However, the last sentence could partially belong to Methods. More importantly, it is not clear (reading the last sentence from the previous paragraph, the study aims) if the overall objective was to explore/analyse communication patterns or to measure the outcomes of an intervention, improving pharmacy service. After reading whole manuscript, a descriptive study was aimed.

Methods

Page 2. An overall study design description, such as the one presented in the previous paragraph, seems to be missing in the beginning of this section.

Page 3, Lines 6-12. Although all staff members invited to collect data were certified on the service provision, it should be stated who they are. The reader is later informed on advisers' training and experience, including the differentiation pharmacist and counter assistant (page 5), but this should be better defined here.

Page 4, Lines 16-20. Since smokers pairing did occur, I was expecting here some more information on smokers' variables, besides a classification criterion.

Page 4, Line 39. Would have been interesting to know the pharmacy remuneration scheme, if by intervention or dependent on outcomes.

Page 4, Lines 53 and 54. It should be clearer how do you control for the effects of adviser's experience and role in your quali data analysis.

Page 5, Lines 3-12. The description here is very complete. An open/natural coding helps the thematic analysis, but sound quali research should present a theoretical frame, which may be from the anthropological or the psychological field. No reference to models, such as the trans-theoretical one, were made, although concepts such as motivation and relapse were mentioned in Findings.

Findings

Overall, I wasn't able to see in Findings an evident lean to the 'voice of medicine' counteracting the motivational interviewing, as mentioned by authors in the Abstract. I couldn't find an initial and the simplest sign of a medicalization attitude from advisers e.g. medicines use (replacement therapies).

Page 7. Table 2 is not possible to read completely. Anyway, it looks like some information here was already presented in Table 1. I'm not sure if both tables are needed, but some tables integration (if possible) might optimized results presentation.

Page 11. I couldn't find in Table 3 the items a), b) and c) mentioned in the table caption. Also, the 2nd sentence seems to be a footnote. Knowing consultations 2 and 4 were much shorter than the initial consultation, it would be informative which codes were predominant in the initial encounter and subsequent ones. Actually, I find strange to aggregate results from different consultation durations when communication and interaction length is a major determinant in consultation behavioural changes.

Page 12. Lines 12-22. It is interesting to have a chance of reading a dialogue excerpt, but it looks like conversation analysis rather than participants' quotations from a quali work. Having a dialogue helps contextualization, but it might also reduce an objective assessment/illustration of a code. This is also the case for the long extract on page 14 (lines 6-19), where authors were focusing on illustrating the smoking expenses issue, but my attention was drawn to smell and eating implications.

Discussion

I agree with the overall argument. However, again, I could not see

	objectively that advisers were using the 'voice of medicine'. Perhaps I'm following an over simplification, but I believe that even after training and certification, most advisers do not possess the right patient communication and counselling skills needed. This overall statement, which is also embedded in Clinical Implications, is presented from a closed/hermetic scientific perspective, difficult to understand the main readership. In this sense, I would recommend a depuration/simplification and greater systematization of the Discussion text. For instance, misalignment emerges at the end of page 14 and again along page 15, plus remitting the reader to complexities that are dealt with in another manuscript (2nd paragraph) – no contribution to a fluent understanding in my opinion. So, the presentation style might be appropriate for a thesis, but in my opinion difficult for a reader from an open access journal. If any changes are to be produced, as a suggestion, authors can think of concrete deliverables to less native readers and prudently augment from there. I would have appreciated to read on authors' opinions in relation to no differences from counter assistants and pharmacists, and implications for pharmaceutical higher education.
--	--

REVIEWER	Jude Robinson University of Liverpool, UK
REVIEW RETURNED	24-Mar-2017

GENERAL COMMENTS	This is an interesting paper on an important topic, as it is essential to be able to assess the quality and nature of the support offered to smokers to understand why some people may quit and others don't and this paper goes some way to addressing this. However I do have concerns - point 2 in the abstract does not make sense, and the abstract uses very exaggerated and convoluted language for concepts that should be expressed simply and clearly. This is an issue throughout this paper, and without a real explanation of phenomenological perspectives terms like 'life world' become meaningless and could be better described. Theoretically this paper is underdeveloped and so I think perhaps the theory should be either be better described or removed altogether. My major concern is the claim that this is an ethnographic study - it is very, very clearly not - and all mention of ethnography should be removed from the entire paper. It is not even strictly a piece of qualitative research as more a content analysis of secondary recorded data which happens to use words. The authors have adopted a highly quantitative approach to analysis, though some, very limited evidence of an engagement with the narrative. There is no proper consideration of ethics in the paper, no indication of when and where the consultations took place, how long they lasted, who else was present etc. and this is vital contextual data that will materially affect the quality of the verbal data obtained. No account either of the advisers - were they young, old, male and female - what ethnicity? - and did you case match them as well as of course this may well have influenced how they 'matched' your respondents. The data presented for a qualitative study are very thin, and would not meet the standards of reporting of qualitative data in good journals. I realise space is limited in this journal, so perhaps ensuring readers understand the key concepts from the content analysis is the task here rather than making claims
--

VERSION 1 – AUTHOR RESPONSE

Reviewer: 1

Reviewer Name: Afonso Cavaco

Thank you for your comments. Our responses are as follows.

Abstract

Overall, the abstract is dense and somehow difficult for non-native English readers.

- We have considerably changed this.

In Methods, I would suggest to add a mix-methods approach. Maybe a theoretical backbone for informing thematic analysis would help the reader to locate how the main themes were developed.

- Thank you for suggesting the need to be clearer about the theoretical background, which is helpful.

As the data were collected using a qualitative approach, and it was the analysis that was mixed, we have amended as follows.

In the abstract:

"Method: 1-3 audio-recorded consultations between an adviser and each pair member over 5-6 weeks were analysed using a mixed method approach. First a content analysis was based on deductive coding drawn from a theme-oriented discourse analysis approach and the Roter Interaction Analysis System (RIAS). Core themes were identified through this quantification to explore in detail the qualitative differences and similarities between quitters and non-quitters."

In the main text:

"Matched pairs

We refer to quitters and non-quitters throughout this paper but in consultations 1 and 2 this is a retrospective designation for eventual quitters and non-quitters. We matched 16 pairs of 4-week quitters and non-quitters (i.e. 32 smokers) on gender, ethnicity, age (according to UK Office for National Statistics [ONS] age bands), and smoking intensity (fewer than 10 cigarettes a day, 10-20 a day, more than 20 a day). This was the total number that could be matched on these variables from the data collected. We had initially aimed for 20 matched pairs but obtained saturation at this lower number and so stopped data collection at this point.

Consultations analysis

We used a mixed methods analysis approach, in which themes were identified deductively and inductively and then quantified. Deductive themes were developed from a list of the key linguistic devices used in theme-oriented discourse analysis,³⁵ which considers how language use constructs professional practice, and the structuring of the medical consultation in the Roter Interaction Analysis System (RIAS),³⁶ which is a widely used method for coding and quantitatively analysing the conversational strategies used structurally in medical consultations. Our approach thus has its origins in psycholinguistics and its purpose is to look at what talk itself is doing as constitutive of social action, rather than as representational of inner psychological states, behaviours, beliefs or attitudes.³⁵ To manage the data as we developed the themes, we used a modification of the qualitative content thematic analysis approach outlined in Clark and Braun³⁷ and Rivas.³⁸ Thus the core team immersed themselves in the data, reading and re-reading each consultation transcription and meeting to discuss these. CR led development of themes from the data, constantly comparing deductive and emerging inductive themes with each other and also comparing data within themes to ensure each one was discrete and representative of the relevant data. CR operationalised the themes, for a second researcher (RS) to code 20% of the data independently. Differences were reconciled by discussion, until a Cohen's kappa statistic of 76% was obtained."

- This also led to deletion of two sentences in second of these two paragraphs in the main text, one because it is included in the first para and the second because it is a less explicit reference to the underlying theory than the statement now made above and hence became redundant.

In Results, and following the methodological description, I would start by presenting first the quali results. If content analysis was done 1st, then I would recommend a different wording in Methods.

- We have presented the results in the order in which they were determined. We have therefore tried to clarify the process in the abstract as above and in the main text as below. As we hope the rewritten abstract makes clear, keeping to the initial order of the findings shows how the 'story' develops, of a core set of strategies that we then explore further. We have amended the methods as below, therefore.

"Once themes were determined, a quantitative summary of these was developed to determine core strategies used by advisers. We compared the frequency of themes between and within successful and unsuccessful quitter consultations to look for similarities and note any differences. We report differences between the matched pairs in the findings in terms of (eventual) quitters versus non-quitters. We counted the number of times a theme was coded overall and divided this by the number of consultations in which it occurred to get an idea of the intensity within a consultation. We also considered the number of smokers for whom each theme was cited, to get an idea of which themes formed a core set for advisers. This process enabled us to focus on themes that were of high or moderate intensity and core usage and associated with particular outcomes, as well as strategies rarely used but that might be important for service improvement. We then explored these themes in more detail, as reported on here. Given the small numbers involved, tests of significance are not reported here."

Findings

There is a missing 1) item, as 2) and 3) exist.

- We have deleted these numbers in the rewriting.

Although sample pairing is mentioned, it was not clear to me if differences were aimed and found.

- We have specified that we looked for differences in the amended paragraph in point 3 above in the main text. The remainder of the paper talks about the differences as quitters vs non-quitters. This is reflected in the paragraph above in the following sentence:

We report differences between the matched pairs in the findings in terms of (eventual) quitters versus non-quitters.

Strengths & Limitations

Page 3, Line 11. Theme saturation is a methodological requirement for baseline quality in qualitative research. I would recommend to drop this from the 2nd bullet point.

- Yes we added it to demonstrate that we met this requirement but have now removed it.

Introduction

Page 2, Lines 43-49. The last paragraph is an important description of the work accomplished.

However, the last sentence could partially belong to Methods. More importantly, it is not clear (reading the last sentence from the previous paragraph, the study aims) if the overall objective was to explore/analyse communication patterns or to measure the outcomes of an intervention, improving pharmacy service. After reading whole manuscript, a descriptive study was aimed.

- We have amended this as follows:

"Moreover, the proportion of smokers enrolling on the Stop Smoking programme in community pharmacies rather than GP surgeries and other settings has remained stable (between 18-21%) since the programme began in 2006 despite government efforts to shift some of the GP workload to them.^{7,16} [sentence removed and added below]

Enhanced services have necessitated a change in mind set for pharmacy staff with a move towards the person-centred care model currently advocated within health services.¹⁷⁻²⁴ Pharmacy staff tend to be inadequately trained in this model,²² and may lack necessary consultation skills,^{22,24} which may affect the growth of the service in pharmacies. We therefore used a 'focused ethnography'^{25,26} approach to explore counselling strategies used in consultations between pharmacy Stop Smoking advisers and smokers. Our primary aim was to use the findings to develop an intervention intended to improve both service uptake and quit rates in the community pharmacy Stop Smoking programme, as reported elsewhere.²⁷"

Methods

Page 2. An overall study design description, such as the one presented in the previous paragraph, seems to be missing in the beginning of this section.

- Yes we wished to avoid repetition, but given the amendments above have now added the following section at the start of the methods. This also answers comments by reviewer 2.

"Focused ethnography

Using focused ethnography^{25,26} we collected and analysed recordings of naturalistic (real world) consultations between pharmacy Stop Smoking advisers and smokers; recordings were made by the advisers themselves for the study. We considered verbal interactions that might affect smoker engagement, maintenance within the pharmacy programme and success in quitting.

In traditional ethnographic work, researchers immerse themselves in the social worlds of their participants to better understand the ways they interact with these worlds. They then generate textual or visual accounts of this for dissemination.²⁸ In focused ethnography, field visits are either much shorter, and selectively focussed, or replaced by audio- or video- or photo-recordings which may be undertaken with the researcher absent from the scene, as in our study. This approach is useful for answering a specific research question when traditional ethnography is impractical.²⁶ "

Page 3, Lines 6-12. Although all staff members invited to collect data were certified on the service provision, it should be stated who they are. The reader is later informed on advisers' training and experience, including the differentiation pharmacist and counter assistant (page 5), but this should be better defined here.

- We have added a phrase as below

"Within pharmacies, we invited any staff members certificated through the National Centre for Smoking Cessation and Training (NCSCT) programme, with a mixture of pharmacists and counter assistants therefore included in the study. We explained the study, took consent, and trained them in research methods and governance, so that they could take smoker..."

Page 4, Lines 16-20. Since smokers pairing did occur, I was expecting here some more information on smokers' variables, besides a classification criterion.

- This is in fact provided in the last paragraph on the same page.

Page 4, Line 39. Would have been interesting to know the pharmacy remuneration scheme, if by intervention or dependent on outcomes.

- We purposely did not put this since its effect on adviser performance is complex, which we consider in a different paper. We have therefore now referenced this as an alternative to adding text:

"3. the consultation four weeks post proposed quit date at which the smoker's quit status is formally designated (as the 4-week quit status) for Public Health England audit statistics and pharmacy remuneration³³ (see Sohanpal et al [2016]³⁴ for a discussion of this in relation to our larger study)."

Page 4, Lines 53 and 54. It should be clearer how do you control for the effects of adviser's experience and role in your quali data analysis.

- We did not collect data that enabled us to consider this as mentioned in the study limitations section. In our published interview-based companion analysis again we did not control for the effects of adviser experience and role. We felt that there were too many other relevant variables that we were not able to consider. In relation to the current analysis, this was focussed on what talk itself does rather than whether different advisers used different strategies and thus this omission is less problematic than if a different form of analysis was used. Of course, in qualitative research it is more important to be transparent about such variables than to control for them. To control for all variables would have required a larger sample size across more pharmacies. Within the limitations of our dataset there was no discernible difference within the consultations but this was not what we were looking at. We have tried to clarify this as follows:

"Importantly, findings were similar whether the adviser was a pharmacist or trained counter assistant. The small number of advisers in the study precluded analysis of the effect of other variables on the consultation. In any case our analysis was focussed on what talk itself does, and was not intended to explore these effects."

- In the limitations section we similarly now refer to this:

"A number of contextual and demographic variables are known to affect pharmacist-smoker communication and quit successes.⁸³ We were unable to explore these in our small sample. We did not case match by adviser ethnicity or other adviser variables as this would have required oversampling for our specific research question, which was not intended to explore this, but this might be a topic for different research. However we systematically looked for adviser-specific patterns and found that our analysis held irrespective of adviser."

Page 5, Lines 3-12. The description here is very complete. An open/natural coding helps the thematic analysis, but sound quali research should present a theoretical frame, which may be from the anthropological or the psychological field. No reference to models, such as the trans-theoretical one, were made, although concepts such as motivation and relapse were mentioned in Findings.

- We have addressed our underpinnings in a previous comment on the abstract. Our approach has its origins in psycholinguistics and its purpose is to look at the discourse per se rather than to use theoretical frameworks about inner psychological states, behaviour change or more general notions about beliefs, attitudes or behaviour. Our approach specifically considers what language itself is doing, not as representational, but as constitutive of social action. We therefore believe the references already provided are sufficient and that this does not mean that our analysis is unsound – we accept that we should have made our underpinnings more explicit, as we now have done, in response to your other comments, so that this is transparent. Motivation and relapse would be accounted for within consideration of the RIAS but we had already also included a reference defining these.

Findings

Overall, I wasn't able to see in Findings an evident lean to the 'voice of medicine' counteracting the motivational interviewing, as mentioned by authors in the Abstract. I couldn't find an initial and the simplest sign of a medicalization attitude from advisers e.g. medicines use (replacement therapies).

- We did not mean to give the impression that the voice of medicine counteracted the motivational

interviewing. We hope our revised abstract is clearer. We believe that our interpretation is correct. For example 88% of quitters and 69% of non-quitters received biomedical info and advice, compared with 63% and 56% involved in lifeworld talk while motivational talk occurred in only 31% of each.

Page 7. Table 2 is not possible to read completely. Anyway, it looks like some information here was already presented in Table 1. I'm not sure if both tables are needed, but some tables integration (if possible) might optimized results presentation.

- We have deleted Table 1.

Page 11. I couldn't find in Table 3 the items a), b) and c) mentioned in the table caption.

- We have made this clearer by specifying the columns and have therefore also described the first two columns to reduce confusion.

Table 3: Quantitative comparisons between quitters and non-quitters in: a) number of quitters and non-quitters for which each code was identified (second and third columns); b) number of consultations overall in which each code was identified (middle columns); c) total number of occurrences in all consultations (penultimate column), f) relative frequency with which codes were used in individual consultations (final column).

Also, the 2nd sentence seems to be a footnote.

- Transformed to a footnote.

Knowing consultations 2 and 4 were much shorter than the initial consultation, it would be informative which codes were predominant in the initial encounter and subsequent ones. Actually, I find strange to aggregate results from different consultation durations when communication and interaction length is a major determinant in consultation behavioural changes.

- We have undertaken a longitudinal analysis of the data and our initial draft included this. However before submission we decided to remove it as it did not add to clarity and nor did it contradict anything we say from the aggregated data analysis; this was particularly because there were fewer recordings for weeks 2 and 4. It must be remembered also that we are looking at what talk does, and we also found that consultation length had no effect on findings, in our initial analysis. We are undertaking a full RIAS analysis to explore such matters in more depth. The first consultation should consider smoker tobacco use and motivation to quit (some advisers used a questionnaire type script) and subsequent ones build on this. Even within this framework, which was upheld, the differences we found were clearly evident. We have tried to explain this with the following text addition in the text just after table 3 (now table 2):

"These were made up primarily of the subthemes developed deductively from psycholinguistics as described in the methods. Table 3 provides a quantitative overview of the themes and selected subthemes, comparing quitters and non-quitters. We follow this with a narrative description of themes; extracts are identified by smoker (P), adviser (A) and consultation week (W) while the quantifiers most, common or many, occasionally or some, and a few or rarely, refer to a theme occurrence of 75%+, 50-74%, 25-49% and less than 25%. The first consultation should consider smoker tobacco use and motivation to quit (some advisers used a questionnaire type script) and subsequent ones build on this. Here we focus on what the talk does within any adviser-smoker interaction and the association of interaction strategies per se with quitting or not, hence aggregating the data whilst mindful of which consultations they represent. We consider the data by consultation sequence in further analyses to be reported elsewhere, which support the findings of this paper."

Page 12. Lines 12-22. It is interesting to have a chance of reading a dialogue excerpt, but it looks like conversation analysis rather than participants' quotations from a quali work. Having a dialogue helps contextualization, but it might also reduce an objective assessment/illustration of a code. This is also

the case for the long extract on page 14 (lines 6-19), where authors were focusing on illustrating the smoking expenses issue, but my attention was drawn to smell and eating implications.

- Yes this is because we did not make our approach clear, as your comments earlier have now enabled us to do. This is in fact, as already stated, an exploration of the data based on psycholinguistics and as we have now made explicit, we were less interested in the points around eating and smell than in how the talk in the interaction itself unfolded. We have considered smell and eating, but in terms of alignment. As we have now made explicit in the methods section, we are not interested so much in the representational nature of talk, ie considering such themes as smell and eating, but in the constitutive nature of talk in shaping social action. We hope that now we have rectified our description, the reviewer is satisfied.

Discussion

I agree with the overall argument. However, again, I could not see objectively that advisers were using the 'voice of medicine'. Perhaps I'm following an over simplification, but I believe that even after training and certification, most advisers do not possess the right patient communication and counselling skills needed. This overall statement, which is also embedded in Clinical Implications, is presented from a closed/hermetic scientific perspective, difficult to understand the main readership.

In this sense, I would recommend a depuration/simplification and greater systematization of the Discussion text. For instance, misalignment emerges at the end of page 14 and again along page 15, plus remitting the reader to complexities that are dealt with in another manuscript (2nd paragraph) – no contribution to a fluent understanding in my opinion.

- We have rewritten the entire discussion in response to these points and hope this is now satisfactory.

So, the presentation style might be appropriate for a thesis, but in my opinion difficult for a reader from an open access journal. If any changes are to be produced, as a suggestion, authors can think of concrete deliverables to less native readers and prudently augment from there.

- It is true that a psycholinguistics approach is less easy to understand than a simple thematic analysis, as we have to use precise words to ensure that the analysis has currency internationally in the approach. However we hope that by responding to the reviewer we have made the paper easier to understand. We feel it is important for such studies to be published in Journals such as BMJ Open as they are increasingly being undertaken in complex intervention design.

I would have appreciated to read on authors' opinions in relation to no differences from counter assistants and pharmacists, and implications for pharmaceutical higher education.

- We are about to publish another paper considering our training programme, developed from the work reported in this paper.

Reviewer: 2

Reviewer Name: Jude Robinson

Institution and Country: University of Liverpool, UK

Please state any competing interests: None declared

Please leave your comments for the authors below

This is an interesting paper on an important topic, as it is essential to be able to assess the quality and nature of the support offered to smokers to understand why some people may quit and others don't and this paper goes some way to addressing this. However I do have concerns - point 2 in the

abstract does not make sense, and the abstract uses very exaggerated and convoluted language for concepts that should be expressed simply and clearly. This is an issue throughout this paper, and without a real explanation of phenomenological perspectives terms like 'life world' become meaningless and could be better described. Theoretically this paper is underdeveloped and so I think perhaps the theory should be either better described or removed altogether.

- We hope that our comments to reviewer 1 have addressed these comments. Thus we are using a psycholinguistics approach which includes our drawing on the Roter Interaction Analysis System - lifeworld talk forms a component of this.

My major concern is the claim that this is an ethnographic study - it is very, very clearly not - and all mention of ethnography should be removed from the entire paper.

- Again we hope that our responses to reviewer 1 address this. Our work very much fits with the concept of 'focused ethnography' which includes the analysis of recorded data. This is certainly not a traditional ethnography and we are sorry for the confusion that has resulted from our not being specific before.

It is not even strictly a piece of qualitative research as more a content analysis of secondary recorded data which happens to use words. The authors have adopted a highly quantitative approach to analysis, though some, very limited evidence of an engagement with the narrative.

- While we did include a quantitative overview of the data, this was simply intended to inform the analysis of the talk itself, which we are confident that we engaged with quite considerably, having also discussed the interpretations and data at length at international meetings. Clearly then our failing has been in making this apparent in the text and also our theoretical underpinnings. Now that it is clear that we have undertaken a focused ethnography, which typically involves collection and analysis of recorded data (which was recorded specifically for this analysis rather than this being secondary analysis) and now that we have made changes that make our approach clearer, we hope our changes have improved the reviewer's view. We ourselves agree that it would be good to include more extracts which would highlight our engagement with these, but were precluded from doing so by the word count. Should the editor wish, we could include these as supplementary material.

There is no proper consideration of ethics in the paper,

- We did cite the ethics approval, and some ethical considerations as below (all in the original submission) and can add to this as needed if the editor requires this.

"Within pharmacies, we invited any staff members certificated through the National Centre for Smoking Cessation and Training (NCSCT) programme, with a mixture of pharmacists and counter assistants therefore included in the study. We explained the study, took consent, and trained them in research methods and governance, so that they could take smoker service user consent and record, securely store and transfer their own consultations to the research team. This meant the natural rhythm of the service was not disrupted. Ethical approval for the study was obtained from the NRES Committee South Central, Berkshire B (reference number 13/SC/0189)."

no indication of when and where the consultations took place, how long they lasted, who else was present etc. and this is vital contextual data that will materially affect the quality of the verbal data obtained.

- You are right, it is good practice to specify such things and in fact we had already done so in the original draft – see the section below which was in the submitted draft. We have added one sentence below to further clarify as your comment was helpful in showing where we could add to this.

"Pharmacies and advisers

Consultations were recorded between November 2013 and May 2014 at community pharmacies contracted by Public Health England to provide the Stop Smoking programme in three inner East London boroughs (Newham, Tower Hamlets, the City and Hackney). These areas are economically

disadvantaged compared with other areas of London and with England as a whole²⁵ and have relatively poor NHS Stop Smoking programme attendance.^{13,15,16} We only included community pharmacy chains or independent single pharmacies, which are distinct from those in institutional settings such as NHS hospitals. Consultations had to take place in a private room. Only the adviser and smoker were present though sometimes the consultations were interrupted as they often took place in rooms that doubled as store rooms, which other staff sometimes needed to access.

.....

Stop Smoking advisers consented eligible smokers and audiorecorded:

1. their first consultation, which tends to set the parameters for the remaining sessions, includes the smoker setting a proximal quit date, and is usually up to three times as long as subsequent sessions (averaging 15 minutes vs 5 minutes);²⁸"

- We decided not to specify the length of the recorded consultations we used as they generally conformed to this pattern but with a large range that bore no relation to eventual quit status or any other variable that we measured. We can add this if the editor sees fit.

No account either of the advisors - were they young, old, male and female - what ethnicity?

- Such detail that was relevant was in fact specified in the following paragraph in the initial submission (with only the last sentences new). We avoided adding so much data that individuals could be identified ie we had to be careful not to break anonymity:

"Participants

Thirty-nine percent (11/28) of recruited advisers provided 158 audiorecordings from 53 smokers. Considering only the 16 matched pairs, these were seen by 9 of the 11 advisers, with a median of three smokers each (range 1-7). Table 1 provides detail on the matched pairs. Whilst women were under-represented in the matched pairs, other characteristics related to cessation outcome were broadly similar. This includes medication use; though we did not monitor for this systematically, the transcripts indicate that similar numbers of quitters and non-quitters were on varenicline for example. Identification (ID) numbers in the tables are also used to attribute extracts from the data in the narrative text. Most advisers in the total group were male (89%) and from pharmacies in City and Hackney (89%). Sixty-seven percent of advisers were Indian, and 11% were from other black and minority ethnic (BME) groups, with missing ethnicity data for one. Importantly, findings were similar whether the adviser was a pharmacist or trained counter assistant. The small number of advisers in the study precluded analysis of the effect of other variables on the consultation. In any case our analysis was focussed on what talk itself does, and was not intended to explore these effects."

and did you case match them as well as of course this may well have influenced how they 'matched' your respondents.

- We felt that with regard to advisers there were too many other relevant variables that we were not able to consider because of the way sampling was undertaken, which was not designed to consider adviser variables. In relation to the current analysis, this was focussed on what talk itself does rather than whether different advisers used different strategies with different demographics and thus this omission is less problematic than if a different form of analysis was used. Of course, in qualitative research it is more important to be transparent about such variables than to control for them. To control for all variables would have required a larger sample size across more pharmacies and our focus was on comparing quitters and non-quitters. Within the limitations of our dataset there was no discernible difference within or between the consultations of advisers based on their characteristics but this was not what we were looking at. We have tried to clarify this as follows:

"Importantly, findings were similar whether the adviser was a pharmacist or trained counter assistant. The small number of advisers in the study precluded analysis of the effect of other variables on the consultation. In any case our analysis was focussed on what talk itself does, and was not intended to explore these effects."

- In the limitations section we similarly now refer to this:

"A number of contextual and demographic variables are known to affect pharmacist-smoker communication and quit successes.⁸³ We were unable to explore these in our small sample. We did not case match by adviser ethnicity or other adviser variables as this would have required oversampling for our specific research question, which was not intended to explore this, but this might be a topic for different research. However we systematically looked for adviser-specific patterns and found that our analysis held irrespective of adviser."

The data presented for a qualitative study are very thin, and would not meet the standards of reporting of qualitative data in good journals. I realise space is limited in this journal, so perhaps ensuring readers understand the key concepts from the content analysis is the task here rather than making claims from one or two data extracts?

- We are sorry you feel this way. We would indeed be happy to provide more extracts but as you say were precluded from doing so by word count. Nonetheless we are in some ways also a little mystified as we have provided sufficient examples for each overarching theme, and not so many that the paper is simply a presentation of the data. If you mean that as a mixed analysis paper the qualitative data are half what they might be if considered alone, then of course this is true, but we feel that the two approaches are complementary and should remain within the one paper. We would remove the quantitative rather than qualitative text should there be the need to choose, and expand on it. Should the editor wish, we could and would be delighted to provide examples by each subtheme, as supplementary material. It may be that the reviewer means that only three overarching themes were explored, but this reflects the nature of the data.

VERSION 2 – REVIEW

REVIEWER	Afonso M Cavaco Faculty of Pharmacy, University of Lisbon
REVIEW RETURNED	14-Jun-2017

GENERAL COMMENTS	As a general comment, I've found this manuscript quite dense overall, particularly in its methodological approach. I wasn't being able to straightforwardly understand all reasons and advantages supporting the study options made e.g. quantifications by percentages and comparisons within a small sample size as well as the use of the (original, modified?) RIAS coding scheme. My other few comments are as follows. Methods – Pharmacies and advisers. The study design comprised paired smokers/non-smokers, with all staff qualified to act as an adviser possible to be enrolled. This seemed not to have had an impact on the quitting rates, but on the counselling process, not controlling for this seems somewhat strange. To which extent were you a priori sure of the non-effect from the perceived level of education and staff authority impact on the exchange? Methods – Matched pairs. It seems to exist a contradiction at the end of this paragraph. In one hand, the aim was to match pairs per 4 variables (with their different levels), clearly reaching more than 20 different pairs, even if the initial aim was 20, as described. On the other hand, the total possible number from data was 16 pairs, and (fortunately) this was enough to reach saturation. Nothing is described if saturation was not reached at the 16th pair. My suggestion is to delete the last but one sentence in this paragraph.
--

	Methods – Consultation analysis. The analytical approach is very thick and detailed. I just wonder if nothing from the focused ethnography framework is informing this stage in qualitative matters i.e. it only served the data collection? Authors also mentioned a modified content thematic analysis and I'm not sure how all this work together, including the linguistic theme-oriented discourse analysis. The last paragraph in page 5 starts with "Once themes were determined,..." which adds uncertainty to what was deductively (i.e. confirmatory) and inductively found. Maybe a graphical/diagrammatic explanation would help most readers. Methods – page 5, lines 46 to 48. I'm not sure if the frequency of themes divided by the number of consultations (where occurring) provides an intensity measure within each consultation. I would prefer to use has denominator the total number of themes in that consultation, find an average for all consultation at the end. Also, I'm not sure if this is what authors are presenting in the last column in Table 2, or if/how this matches with lines 27 and 28 in page 7 (Findings). Actually, I was unable to see the usefulness of this specific quantification while reading Findings and Discussion. Findings – page 12, lines 13 to 16. Authors have provided contextualized transcripts, which is an important option. However, I'm not sure if they are always clearly supporting the themes. For instance, this dialogue excerpt shows an initial misalignment from the adviser, but soon to be corrected from the exchange process point of view, i.e. a confirmatory closed question was used and the adviser moved on to identify the reasons for trying to quit smoking (without falling or insisting in the previous assumptions). Strengths and limitations – Authors mentioned the lack of matching through adviser variables. Nevertheless, nothing was stated concerning pharmacists and trained counter assistants having the same performance results when power and (hopefully professional) authority differentials were addressed and should be obvious amongst community pharmacies staff. It was disappointing not to read any remarks on the lack of preparation for effective smoke quitting success, knowing the higher healthcare education background for pharmacists. Clinical implications – It was surprising the authors only focused their comments on the poor performance of the advisers, not mentioning the probable misalignment of the training programme to community pharmacy practitioners' education and/or practice skills. Otherwise, authors should provide evidence on how this programme has been effective with other healthcare professionals.
--	--

VERSION 2 – AUTHOR RESPONSE

Reviewer: 1

Reviewer Name: Afonso Cavaco

Thank you for your comments. Our responses are as follows.

Abstract

Overall, the abstract is dense and somehow difficult for non-native English readers.

- We have considerably changed this.

In Methods, I would suggest to add a mix-methods approach. Maybe a theoretical backbone for informing thematic analysis would help the reader to locate how the main themes were developed.

- Thank you for suggesting the need to be clearer about the theoretical background, which is helpful.

As the data were collected using a qualitative approach, and it was the analysis that was mixed, we have amended as follows.

In the abstract:

"Method: 1-3 audio-recorded consultations between an adviser and each pair member over 5-6 weeks were analysed using a mixed method approach. First a content analysis was based on deductive coding drawn from a theme-oriented discourse analysis approach and the Roter Interaction Analysis System (RIAS). Core themes were identified through this quantification to explore in detail the qualitative differences and similarities between quitters and non-quitters."

In the main text:

"Matched pairs

We refer to quitters and non-quitters throughout this paper but in consultations 1 and 2 this is a retrospective designation for eventual quitters and non-quitters. We matched 16 pairs of 4-week quitters and non-quitters (i.e. 32 smokers) on gender, ethnicity, age (according to UK Office for National Statistics [ONS] age bands), and smoking intensity (fewer than 10 cigarettes a day, 10-20 a day, more than 20 a day). This was the total number that could be matched on these variables from the data collected. We had initially aimed for 20 matched pairs but obtained saturation at this lower number and so stopped data collection at this point.

Consultations analysis

We used a mixed methods analysis approach, in which themes were identified deductively and inductively and then quantified. Deductive themes were developed from a list of the key linguistic devices used in theme-oriented discourse analysis,³⁵ which considers how language use constructs professional practice, and the structuring of the medical consultation in the Roter Interaction Analysis System (RIAS),³⁶ which is a widely used method for coding and quantitatively analysing the conversational strategies used structurally in medical consultations. Our approach thus has its origins in psycholinguistics and its purpose is to look at what talk itself is doing as constitutive of social action, rather than as representational of inner psychological states, behaviours, beliefs or attitudes.³⁵ To manage the data as we developed the themes, we used a modification of the qualitative content thematic analysis approach outlined in Clark and Braun³⁷ and Rivas.³⁸ Thus the core team immersed themselves in the data, reading and re-reading each consultation transcription and meeting to discuss these. CR led development of themes from the data, constantly comparing deductive and emerging inductive themes with each other and also comparing data within themes to ensure each one was discrete and representative of the relevant data. CR operationalised the themes, for a second researcher (RS) to code 20% of the data independently. Differences were reconciled by discussion, until a Cohen's kappa statistic of 76% was obtained."

- This also led to deletion of two sentences in second of these two paragraphs in the main text, one because it is included in the first para and the second because it is a less explicit reference to the underlying theory than the statement now made above and hence became redundant.

In Results, and following the methodological description, I would start by presenting first the quali results. If content analysis was done 1st, then I would recommend a different wording in Methods.

- We have presented the results in the order in which they were determined. We have therefore tried to clarify the process in the abstract as above and in the main text as below. As we hope the rewritten abstract makes clear, keeping to the initial order of the findings shows how the 'story' develops, of a core set of strategies that we then explore further. We have amended the methods as below, therefore.

"Once themes were determined, a quantitative summary of these was developed to determine core strategies used by advisers. We compared the frequency of themes between and within successful

and unsuccessful quitter consultations to look for similarities and note any differences. We report differences between the matched pairs in the findings in terms of (eventual) quitters versus non-quitters. We counted the number of times a theme was coded overall and divided this by the number of consultations in which it occurred to get an idea of the intensity within a consultation. We also considered the number of smokers for whom each theme was cited, to get an idea of which themes formed a core set for advisers. This process enabled us to focus on themes that were of high or moderate intensity and core usage and associated with particular outcomes, as well as strategies rarely used but that might be important for service improvement. We then explored these themes in more detail, as reported on here. Given the small numbers involved, tests of significance are not reported here."

Findings

There is a missing 1) item, as 2) and 3) exist.

- We have deleted these numbers in the rewriting.

Although sample pairing is mentioned, it was not clear to me if differences were aimed and found.

- We have specified that we looked for differences in the amended paragraph in point 3 above in the main text. The remainder of the paper talks about the differences as quitters vs non-quitters. This is reflected in the paragraph above in the following sentence:

We report differences between the matched pairs in the findings in terms of (eventual) quitters versus non-quitters.

Strengths & Limitations

Page 3, Line 11. Theme saturation is a methodological requirement for baseline quality in qualitative research. I would recommend to drop this from the 2nd bullet point.

- Yes we added it to demonstrate that we met this requirement but have now removed it.

Introduction

Page 2, Lines 43-49. The last paragraph is an important description of the work accomplished. However, the last sentence could partially belong to Methods. More importantly, it is not clear (reading the last sentence from the previous paragraph, the study aims) if the overall objective was to explore/analyse communication patterns or to measure the outcomes of an intervention, improving pharmacy service. After reading whole manuscript, a descriptive study was aimed.

- We have amended this as follows:

"Moreover, the proportion of smokers enrolling on the Stop Smoking programme in community pharmacies rather than GP surgeries and other settings has remained stable (between 18-21%) since the programme began in 2006 despite government efforts to shift some of the GP workload to them.^{7,16} [sentence removed and added below]

Enhanced services have necessitated a change in mind set for pharmacy staff with a move towards the person-centred care model currently advocated within health services.¹⁷⁻²⁴ Pharmacy staff tend to be inadequately trained in this model,²² and may lack necessary consultation skills,^{22,24} which may affect the growth of the service in pharmacies. We therefore used a 'focused ethnography'^{25,26} approach to explore counselling strategies used in consultations between pharmacy Stop Smoking advisers and smokers. Our primary aim was to use the findings to develop an intervention intended to improve both service uptake and quit rates in the community pharmacy Stop Smoking programme, as reported elsewhere.²⁷"

Methods

Page 2. An overall study design description, such as the one presented in the previous paragraph,

seems to be missing in the beginning of this section.

- Yes we wished to avoid repetition, but given the amendments above have now added the following section at the start of the methods. This also answers comments by reviewer 2.

"Focused ethnography

Using focused ethnography^{25,26} we collected and analysed recordings of naturalistic (real world) consultations between pharmacy Stop Smoking advisers and smokers; recordings were made by the advisers themselves for the study. We considered verbal interactions that might affect smoker engagement, maintenance within the pharmacy programme and success in quitting.

In traditional ethnographic work, researchers immerse themselves in the social worlds of their participants to better understand the ways they interact with these worlds. They then generate textual or visual accounts of this for dissemination.²⁸ In focused ethnography, field visits are either much shorter, and selectively focussed, or replaced by audio- or video- or photo-recordings which may be undertaken with the researcher absent from the scene, as in our study. This approach is useful for answering a specific research question when traditional ethnography is impractical.²⁶ "

Page 3, Lines 6-12. Although all staff members invited to collect data were certified on the service provision, it should be stated who they are. The reader is later informed on advisers' training and experience, including the differentiation pharmacist and counter assistant (page 5), but this should be better defined here.

- We have added a phrase as below

"Within pharmacies, we invited any staff members certificated through the National Centre for Smoking Cessation and Training (NCSCT) programme, with a mixture of pharmacists and counter assistants therefore included in the study. We explained the study, took consent, and trained them in research methods and governance, so that they could take smoker..."

Page 4, Lines 16-20. Since smokers pairing did occur, I was expecting here some more information on smokers' variables, besides a classification criterion.

- This is in fact provided in the last paragraph on the same page.

Page 4, Line 39. Would have been interesting to know the pharmacy remuneration scheme, if by intervention or dependent on outcomes.

- We purposely did not put this since its effect on adviser performance is complex, which we consider in a different paper. We have therefore now referenced this as an alternative to adding text:

"3. the consultation four weeks post proposed quit date at which the smoker's quit status is formally designated (as the 4-week quit status) for Public Health England audit statistics and pharmacy remuneration³³ (see Sohanpal et al [2016]³⁴ for a discussion of this in relation to our larger study)."

Page 4, Lines 53 and 54. It should be clearer how do you control for the effects of adviser's experience and role in your quali data analysis.

- We did not collect data that enabled us to consider this as mentioned in the study limitations section.

In our published interview-based companion analysis again we did not control for the effects of adviser experience and role. We felt that there were too many other relevant variables that we were not able to consider. In relation to the current analysis, this was focussed on what talk itself does rather than whether different advisers used different strategies and thus this omission is less problematic than if a different form of analysis was used. Of course, in qualitative research it is more important to be transparent about such variables than to control for them. To control for all variables would have required a larger sample size across more pharmacies. Within the limitations of our dataset there was no discernible difference within the consultations but this was not what we were

looking at. We have tried to clarify this as follows:

"Importantly, findings were similar whether the adviser was a pharmacist or trained counter assistant. The small number of advisers in the study precluded analysis of the effect of other variables on the consultation. In any case our analysis was focussed on what talk itself does, and was not intended to explore these effects."

- In the limitations section we similarly now refer to this:

"A number of contextual and demographic variables are known to affect pharmacist-smoker communication and quit successes.⁸³ We were unable to explore these in our small sample. We did not case match by adviser ethnicity or other adviser variables as this would have required oversampling for our specific research question, which was not intended to explore this, but this might be a topic for different research. However we systematically looked for adviser-specific patterns and found that our analysis held irrespective of adviser."

Page 5, Lines 3-12. The description here is very complete. An open/natural coding helps the thematic analysis, but sound quali research should present a theoretical frame, which may be from the anthropological or the psychological field. No reference to models, such as the trans-theoretical one, were made, although concepts such as motivation and relapse were mentioned in Findings.

- We have addressed our underpinnings in a previous comment on the abstract. Our approach has its origins in psycholinguistics and its purpose is to look at the discourse per se rather than to use theoretical frameworks about inner psychological states, behaviour change or more general notions about beliefs, attitudes or behaviour. Our approach specifically considers what language itself is doing, not as representational, but as constitutive of social action. We therefore believe the references already provided are sufficient and that this does not mean that our analysis is unsound – we accept that we should have made our underpinnings more explicit, as we now have done, in response to your other comments, so that this is transparent. Motivation and relapse would be accounted for within consideration of the RIAS but we had already also included a reference defining these.

Findings

Overall, I wasn't able to see in Findings an evident lean to the 'voice of medicine' counteracting the motivational interviewing, as mentioned by authors in the Abstract. I couldn't find an initial and the simplest sign of a medicalization attitude from advisers e.g. medicines use (replacement therapies).

- We did not mean to give the impression that the voice of medicine counteracted the motivational interviewing. We hope our revised abstract is clearer. We believe that our interpretation is correct. For example 88% of quitters and 69% of non-quitters received biomedical info and advice, compared with 63% and 56% involved in lifeworld talk while motivational talk occurred in only 31% of each.

Page 7. Table 2 is not possible to read completely. Anyway, it looks like some information here was already presented in Table 1. I'm not sure if both tables are needed, but some tables integration (if possible) might optimized results presentation.

- We have deleted Table 1.

Page 11. I couldn't find in Table 3 the items a), b) and c) mentioned in the table caption.

- We have made this clearer by specifying the columns and have therefore also described the first two columns to reduce confusion.

Table 3: Quantitative comparisons between quitters and non-quitters in: a) number of quitters and non-quitters for which each code was identified (second and third columns); b) number of

consultations overall in which each code was identified (middle columns); c) total number of occurrences in all consultations (penultimate column), f) relative frequency with which codes were used in individual consultations (final column).

Also, the 2nd sentence seems to be a footnote.

- Transformed to a footnote.

Knowing consultations 2 and 4 were much shorter than the initial consultation, it would be informative which codes were predominant in the initial encounter and subsequent ones. Actually, I find strange to aggregate results from different consultation durations when communication and interaction length is a major determinant in consultation behavioural changes.

- We have undertaken a longitudinal analysis of the data and our initial draft included this. However before submission we decided to remove it as it did not add to clarity and nor did it contradict anything we say from the aggregated data analysis; this was particularly because there were fewer recordings for weeks 2 and 4. It must be remembered also that we are looking at what talk does, and we also found that consultation length had no effect on findings, in our initial analysis. We are undertaking a full RIAS analysis to explore such matters in more depth. The first consultation should consider smoker tobacco use and motivation to quit (some advisers used a questionnaire type script) and subsequent ones build on this. Even within this framework, which was upheld, the differences we found were clearly evident. We have tried to explain this with the following text addition in the text just after table 3 (now table 2):

"These were made up primarily of the subthemes developed deductively from psycholinguistics as described in the methods. Table 3 provides a quantitative overview of the themes and selected subthemes, comparing quitters and non-quitters. We follow this with a narrative description of themes; extracts are identified by smoker (P), adviser (A) and consultation week (W) while the quantifiers most, common or many, occasionally or some, and a few or rarely, refer to a theme occurrence of 75%+, 50-74%, 25-49% and less than 25%. The first consultation should consider smoker tobacco use and motivation to quit (some advisers used a questionnaire type script) and subsequent ones build on this. Here we focus on what the talk does within any adviser-smoker interaction and the association of interaction strategies per se with quitting or not, hence aggregating the data whilst mindful of which consultations they represent. We consider the data by consultation sequence in further analyses to be reported elsewhere, which support the findings of this paper."

Page 12. Lines 12-22. It is interesting to have a chance of reading a dialogue excerpt, but it looks like conversation analysis rather than participants' quotations from a quali work. Having a dialogue helps contextualization, but it might also reduce an objective assessment/illustration of a code. This is also the case for the long extract on page 14 (lines 6-19), where authors were focusing on illustrating the smoking expenses issue, but my attention was drawn to smell and eating implications.

- Yes this is because we did not make our approach clear, as your comments earlier have now enabled us to do. This is in fact, as already stated, an exploration of the data based on psycholinguistics and as we have now made explicit, we were less interested in the points around eating and smell than in how the talk in the interaction itself unfolded. We have considered smell and eating, but in terms of alignment. As we have now made explicit in the methods section, we are not interested so much in the representational nature of talk, ie considering such themes as smell and eating, but in the constitutive nature of talk in shaping social action. We hope that now we have rectified our description, the reviewer is satisfied.

Discussion

I agree with the overall argument. However, again, I could not see objectively that advisers were using the 'voice of medicine'. Perhaps I'm following an over simplification, but I believe that even after training and certification, most advisers do not possess the right patient communication and

counselling skills needed. This overall statement, which is also embedded in Clinical Implications, is presented from a closed/hermetic scientific perspective, difficult to understand the main readership.

In this sense, I would recommend a depuration/simplification and greater systematization of the Discussion text. For instance, misalignment emerges at the end of page 14 and again along page 15, plus remitting the reader to complexities that are dealt with in another manuscript (2nd paragraph) – no contribution to a fluent understanding in my opinion.

- We have rewritten the entire discussion in response to these points and hope this is now satisfactory.

So, the presentation style might be appropriate for a thesis, but in my opinion difficult for a reader from an open access journal. If any changes are to be produced, as a suggestion, authors can think of concrete deliverables to less native readers and prudently augment from there.

- It is true that a psycholinguistics approach is less easy to understand than a simple thematic analysis, as we have to use precise words to ensure that the analysis has currency internationally in the approach. However we hope that by responding to the reviewer we have made the paper easier to understand. We feel it is important for such studies to be published in Journals such as BMJ Open as they are increasingly being undertaken in complex intervention design.

I would have appreciated to read on authors' opinions in relation to no differences from counter assistants and pharmacists, and implications for pharmaceutical higher education.

- We are about to publish another paper considering our training programme, developed from the work reported in this paper.

Reviewer: 2

Reviewer Name: Jude Robinson

Institution and Country: University of Liverpool, UK

Please state any competing interests: None declared

Please leave your comments for the authors below

This is an interesting paper on an important topic, as it is essential to be able to assess the quality and nature of the support offered to smokers to understand why some people may quit and others don't and this paper goes some way to addressing this. However I do have concerns - point 2 in the abstract does not make sense, and the abstract uses very exaggerated and convoluted language for concepts that should be expressed simply and clearly. This is an issue throughout this paper, and without a real explanation of phenomenological perspectives terms like 'life world' become meaningless and could be better described. Theoretically this paper is underdeveloped and so I think perhaps the theory should be either be better described or removed altogether.

- We hope that our comments to reviewer 1 have addressed these comments. Thus we are using a psycholinguistics approach which includes our drawing on the Roter Interaction Analysis System - lifeworld talk forms a component of this.

My major concern is the claim that this is an ethnographic study - it is very, very clearly not - and all mention of ethnography should be removed from the entire paper.

- Again we hope that our responses to reviewer 1 address this. Our work very much fits with the concept of 'focused ethnography' which includes the analysis of recorded data. This is certainly not a traditional ethnography and we are sorry for the confusion that has resulted from our not being specific before.

It is not even strictly a piece of qualitative research as more a content analysis of secondary recorded data which happens to use words. The authors have adopted a highly quantitative approach to analysis, though some, very limited evidence of an engagement with the narrative.

- While we did include a quantitative overview of the data, this was simply intended to inform the analysis of the talk itself, which we are confident that we engaged with quite considerably, having also discussed the interpretations and data at length at international meetings. Clearly then our failing has been in making this apparent in the text and also our theoretical underpinnings. Now that it is clear that we have undertaken a focused ethnography, which typically involves collection and analysis of recorded data (which was recorded specifically for this analysis rather than this being secondary analysis) and now that we have made changes that make our approach clearer, we hope our changes have improved the reviewer's view. We ourselves agree that it would be good to include more extracts which would highlight our engagement with these, but were precluded from doing so by the word count. Should the editor wish, we could include these as supplementary material.

There is no proper consideration of ethics in the paper,

- We did cite the ethics approval, and some ethical considerations as below (all in the original submission) and can add to this as needed if the editor requires this.

"Within pharmacies, we invited any staff members certificated through the National Centre for Smoking Cessation and Training (NCSCT) programme, with a mixture of pharmacists and counter assistants therefore included in the study. We explained the study, took consent, and trained them in research methods and governance, so that they could take smoker service user consent and record, securely store and transfer their own consultations to the research team. This meant the natural rhythm of the service was not disrupted. Ethical approval for the study was obtained from the NRES Committee South Central, Berkshire B (reference number 13/SC/0189)."

no indication of when and where the consultations took place, how long they lasted, who else was present etc. and this is vital contextual data that will materially affect the quality of the verbal data obtained.

- You are right, it is good practice to specify such things and in fact we had already done so in the original draft – see the section below which was in the submitted draft. We have added one sentence below to further clarify as your comment was helpful in showing where we could add to this.

"Pharmacies and advisers

Consultations were recorded between November 2013 and May 2014 at community pharmacies contracted by Public Health England to provide the Stop Smoking programme in three inner East London boroughs (Newham, Tower Hamlets, the City and Hackney). These areas are economically disadvantaged compared with other areas of London and with England as a whole²⁵ and have relatively poor NHS Stop Smoking programme attendance.^{13,15,16} We only included community pharmacy chains or independent single pharmacies, which are distinct from those in institutional settings such as NHS hospitals. Consultations had to take place in a private room. Only the adviser and smoker were present though sometimes the consultations were interrupted as they often took place in rooms that doubled as store rooms, which other staff sometimes needed to access.

.....

Stop Smoking advisers consented eligible smokers and audiorecorded:

1. their first consultation, which tends to set the parameters for the remaining sessions, includes the smoker setting a proximal quit date, and is usually up to three times as long as subsequent sessions (averaging 15 minutes vs 5 minutes);²⁸

- We decided not to specify the length of the recorded consultations we used as they generally conformed to this pattern but with a large range that bore no relation to eventual quit status or any other variable that we measured. We can add this if the editor sees fit.

No account either of the advisors - were they young, old, male and female - what ethnicity?
- Such detail that was relevant was in fact specified in the following paragraph in the initial submission (with only the last sentences new). We avoided adding so much data that individuals could be identified ie we had to be careful not to break anonymity:

"Participants

Thirty-nine percent (11/28) of recruited advisers provided 158 audiorecordings from 53 smokers. Considering only the 16 matched pairs, these were seen by 9 of the 11 advisers, with a median of three smokers each (range 1-7). Table 1 provides detail on the matched pairs. Whilst women were under-represented in the matched pairs, other characteristics related to cessation outcome were broadly similar. This includes medication use; though we did not monitor for this systematically, the transcripts indicate that similar numbers of quitters and non-quitters were on varenicline for example. Identification (ID) numbers in the tables are also used to attribute extracts from the data in the narrative text. Most advisers in the total group were male (89%) and from pharmacies in City and Hackney (89%). Sixty-seven percent of advisers were Indian, and 11% were from other black and minority ethnic (BME) groups, with missing ethnicity data for one. Importantly, findings were similar whether the adviser was a pharmacist or trained counter assistant. The small number of advisers in the study precluded analysis of the effect of other variables on the consultation. In any case our analysis was focussed on what talk itself does, and was not intended to explore these effects."

and did you case match them as well as of course this may well have influenced how they 'matched' your respondents.

- We felt that with regard to advisers there were too many other relevant variables that we were not able to consider because of the way sampling was undertaken, which was not designed to consider adviser variables. In relation to the current analysis, this was focussed on what talk itself does rather than whether different advisers used different strategies with different demographics and thus this omission is less problematic than if a different form of analysis was used. Of course, in qualitative research it is more important to be transparent about such variables than to control for them. To control for all variables would have required a larger sample size across more pharmacies and our focus was on comparing quitters and non-quitters. Within the limitations of our dataset there was no discernible difference within or between the consultations of advisers based on their characteristics but this was not what we were looking at. We have tried to clarify this as follows:

"Importantly, findings were similar whether the adviser was a pharmacist or trained counter assistant. The small number of advisers in the study precluded analysis of the effect of other variables on the consultation. In any case our analysis was focussed on what talk itself does, and was not intended to explore these effects."

- In the limitations section we similarly now refer to this:

"A number of contextual and demographic variables are known to affect pharmacist-smoker communication and quit successes.⁸³ We were unable to explore these in our small sample. We did not case match by adviser ethnicity or other adviser variables as this would have required oversampling for our specific research question, which was not intended to explore this, but this might be a topic for different research. However we systematically looked for adviser-specific patterns and found that our analysis held irrespective of adviser."

The data presented for a qualitative study are very thin, and would not meet the standards of reporting of qualitative data in good journals. I realise space is limited in this journal, so perhaps ensuring readers understand the key concepts from the content analysis is the task here rather than making claims from one or two data extracts?

- We are sorry you feel this way. We would indeed be happy to provide more extracts but as you say

were precluded from doing so by word count. Nonetheless we are in some ways also a little mystified as we have provided sufficient examples for each overarching theme, and not so many that the paper is simply a presentation of the data. If you mean that as a mixed analysis paper the qualitative data are half what they might be if considered alone, then of course this is true, but we feel that the two approaches are complementary and should remain within the one paper. We would remove the quantitative rather than qualitative text should there be the need to choose, and expand on it. Should the editor wish, we could and would be delighted to provide examples by each subtheme, as supplementary material. It may be that the reviewer means that only three overarching themes were explored, but this reflects the nature of the data.